# S2abEL: A Dataset for Entity Linking from Scientific Tables

**Yuze Lou**[1] [*] **Bailey Kuehl**[2]**, Erin Bransom**[2]**,**
**Sergey Feldman**[2]**, Aakanksha Naik**[2]**, Doug Downey**[2]
[1]Univserty of Michigan, [2]Allen Institute for AI
yuzelou@umich.edu, {baileyk,erinbransom,sergey,aakankshan,dougd}@allenai.org

## Abstract

Entity linking (EL) is the task of linking a textual mention to its corresponding entry in a knowledge base, and is critical for many knowledge-intensive NLP applications. When applied to tables in scientific papers, EL is a step toward large-scale scientific knowledge bases that could enable advanced scientific question answering and analytics. We present the first dataset for EL in scientific tables. EL for scientific tables is especially challenging because scientific knowledge bases can be very incomplete, and disambiguating table mentions typically requires understanding the paper's text in addition to the table. Our dataset, Scientific Table Entity Linking (S2abEL), focuses on EL in machine learning results tables and includes hand-labeled cell types, attributed sources, and entity links from the PaperswithCode taxonomy for 8,429 cells from 732 tables. We introduce a neural baseline method designed for EL on scientific tables containing many out-of-knowledge-base mentions, and show that it significantly outperforms a state-of-the-art generic table EL method. The best baselines fall below human performance, and our analysis highlights avenues for improvement. Code and the dataset is available at: `https://github.com/allenai/S2abEL/tree/main`.

## 1   Introduction

Entity Linking (EL) is a longstanding problem in natural language processing and information extraction. The goal of the task is to link textual mentions to their corresponding entities in a knowledge base (KB) (Cucerzan, 2007), and it serves as a building block for various knowledge-intensive applications, including search engines (Blanco et al., 2015), question-answering systems (Dubey et al., 2018), and more. However, existing EL methods and datasets primarily focus on linking mentions from free-form natural language (Gu et al., 2021; De Cao et al., 2021; Li et al., 2020; Yamada et al., 2022). Some consider tabular data, but focus on tables from the general domain (Deng et al., 2020; Tang et al., 2021b; Iida et al., 2021; Yu et al., 2019). Despite significant research in EL, there is a lack of datasets and methods for EL in *scientific tables*. Linking entities in scientific tables holds promise for accelerating science in multiple ways: from augmented reading applications that help users understand the meaning of table cells without diving into the document (Head et al., 2021) to automated knowledge base construction that unifies disparate tables, enabling complex question answering or hypothesis generation (Hope et al., 2022).

EL in science is challenging because the set of scientific entities is vast and always growing, and existing knowledge bases are highly incomplete. A traditional "closed world" assumption often made in EL systems, whereby all mentions have corresponding entities in the target KB, is not realistic in scientific domains. It is important to detect which mentions are entities not yet in the reference KB, referred to as *outKB* mentions. Even for human annotators, accurately identifying whether a rarely-seen surface form actually refers to a rarely-mentioned long-tail inKB entity or an outKB entity requires domain expertise and a significant effort to investigate the document and the target KB. A further challenge is that entity mentions in scientific tables are often abbreviated and opaque, and require examining other context in the caption and paper text for disambiguation. An example is shown in Figure 1.

In this paper, we make three main contributions. First, we introduce *S2abEL*, a high-quality human-annotated dataset for EL in machine learning results tables. The dataset is sufficiently large for training and evaluating models on table EL and relevant sub-tasks, including 52,257 annotations of appropriate types for table cells (e.g. method, dataset),

---

[*]Work done during an internship at AI2.

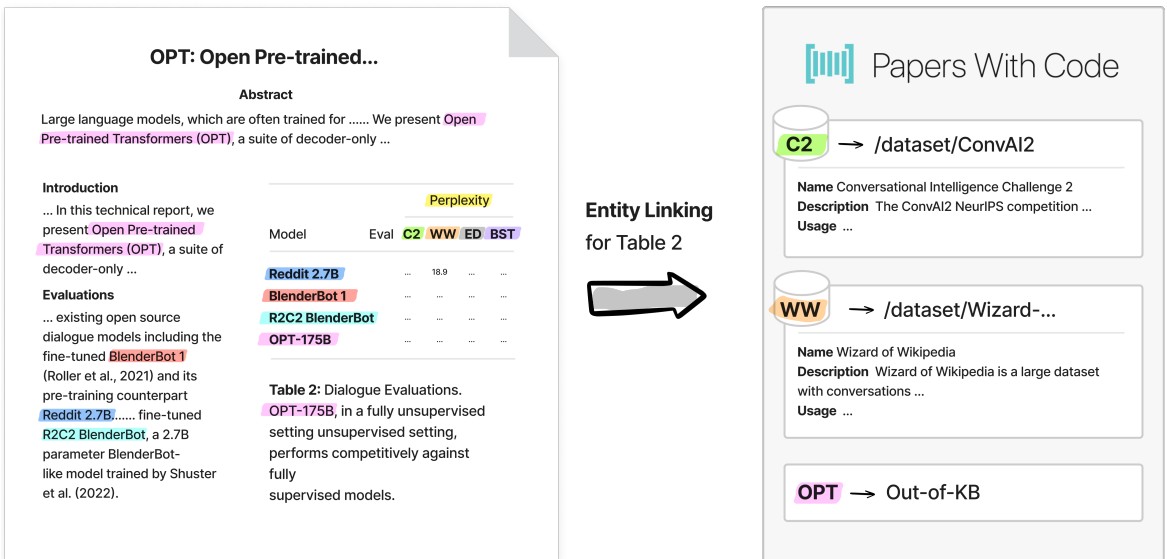

Figure 1: Part of a table in *OPT: Open Pre-trained Transformer Language Models* (Zhang et al., 2022) showing relevant context that must be found for entity mentions in the table, and a portion of EL results mapping table mentions to the PapersWithCode KB.

9,565 annotations of attributed source papers and candidate entities for mentions, and 8,429 annotations for entity disambiguation including outKB mentions. To the best of our knowledge, this is the first dataset for table EL in the scientific domain. Second, we propose a model that serves as a strong baseline for each of the sub-tasks, as well as end-to-end table EL. We conduct a comprehensive comparison between our approach and existing approaches, where applicable, for each sub-task. Our method significantly outperforms TURL (Deng et al., 2020), a state-of-the-art method closest to the table EL task, but only designed for general-domain tables. We also provide a detailed error analysis that emphasizes the need for improved methods to address the unique challenges of EL from scientific tables with outKB mentions.

## 2 Related Work

### 2.1 Entity Linking

In recent years, various approaches have been proposed for entity linking from free-form text, leveraging large language models (Gu et al., 2021; De Cao et al., 2021; Li et al., 2020; Yamada et al., 2019). Researchers have also attempted to extend EL to structured Web tables, but they solely rely on table contents and do not have rich surrounding text (Deng et al., 2020; Zhang et al., 2020; Bhagavatula et al., 2015; Mulwad et al., 2023; Tang et al., 2020; Iida et al., 2021). Most of these works focus

on general-purpose KBs such as Wikidata (Vrandečić and Krötzsch, 2014) and DBPedia (Auer et al., 2007) and typically test their approaches with the assumption that the target KB is complete with respect to the mentions being linked (e.g., De Cao et al., 2021; Deng et al., 2020; Hoffart et al., 2011; Tang et al., 2021a; Yamada et al., 2019).

There is a lack of high-quality datasets for table EL in the scientific domain with abundant outKB mentions. Recent work by Ruas and Couto (2022) provides a dataset that artificially mimics an incomplete KB for biomedical text by removing actual referent entities but linking concepts to the direct ancestor of the referent entities. In contrast, our work provides human-annotated labels of realistic missing entities for scientific tables, without relying on the target KB to contain ancestor relations. Our dataset offers two distinct advantages: first, it provides context from documents in addition to original table mentions, and second, it explicitly identifies mentions referring to outKB entities.

### 2.2 Scientific IE

The field of scientific information extraction (IE) aims to extract structured information from scientific documents. Various extraction tasks have been studied in this area, such as detecting and classifying semantic relations (Jain et al., 2020; Sahu et al., 2016), concept extraction (Fu et al., 2020), automatic leaderboard construction (Kardas et al., 2020; Hou et al., 2019), and citation analysis (Jur-

gens et al., 2018; Cohan et al., 2019).

Among these, Kardas et al., 2020; Hou et al., 2019; Yu et al., 2019, 2020 are the closest to ours. Given a set of papers, they begin by manually extracting a taxonomy of tasks, datasets, and metric names from those papers. Whereas our data set maps each entity to an existing canonical external KB (PwC), they target a taxonomy manually built from the particular papers and surface forms they extract from. Notably, this taxonomy emphasizes lexical representations, with entities such as "AP" and "Average Precision" treated as distinct entities in the taxonomy despite being identical metrics in reality, due to different surface forms appearing in the papers. Such incomplete and ambiguous entity identification makes it difficult for users to interpret the results and limits the practical applicability of the extracted information. In contrast, we propose a dataset and baselines for the end-to-end table EL task, beginning with a table in the context of a paper and ending with each cell linked to entities in the canonicalized ontology of the target KB (or classified as outKB).

## 3 Entity Linking in Scientific Tables

Our entity linking task takes as input a reference KB (the *Papers with Code*[1] taxonomy in our experiments), a table in a scientific paper, and the table's surrounding context. The goal is to output an entity from the KB for each table cell (or "outKB" if none). We decompose the task into several subtasks, discussed below. We then present S2abEL, the dataset we construct for scientific table EL.

### 3.1 Task Definition

**Cell Type Classification (CTC)** is the task of identifying types of entities contained in a table cell, based on the document in which the cell appears. This step is helpful to focus the later linking task on the correct type of entities from the target KB, and also excludes non-entity cells (e.g. those containing numeric values used to report experimental results) from later processing. Such exclusion removes a substantial fraction of table cells (74% in our dataset), reducing the computational cost.

One approach to CTC is to view it as a multi-label classification task since a cell may contain multiple entities of different types. However, our initial investigation found that only mentions of datasets and metrics co-appear to a notable degree

[1] https://paperswithcode.com/

(e.g., "QNLI (acc)" indicates the *accuracy* of some method evaluated on the *Question-answering NLI* dataset (Wang et al., 2018)). Therefore, we introduce a separate class for these instances, reducing CTC to a **single-label** classification task with four positive classes: *method*, *dataset*, *metric*, and *dataset&metric*.

**Attributed Source Matching (ASM)** is the task of identifying *attributed source(s)* for a table cell within the context of the document. The *attributed source(s)* for a concept in a document $p$ is the reference paper mentioned in $p$ to which the authors of $p$ attribute the concept. ASM is a crucial step in distinguishing similar surface forms and finding the correct referent entities. For example, in Figure 1, ASM can help clarify which entities "BlenderBot 1" and "R2C2 BlenderBot" refer to, as the first mention is attributed to Roller et al., 2021 while the second mention is attributed to Shuster et al., 2022. Identifying these attributions helps a system uniquely identify these two entities despite their very similar surface forms and the fact that their contexts in the document often overlap. In this work, we consider the documents listed in the reference section and the document itself as potential sources for attribution. The inclusion of the document itself is necessary since concepts may be introduced in the current document for the first time.

**Candidate Entity Retrieval (CER)** is the process of identifying a small set of entities from the target KB that are most likely to be the referent entity for a table cell within the context of the document. The purpose of this step is to exclude unlikely candidates and pass only a limited number of candidates to the next step, to reduce computational cost.

**Entity Disambiguation (ED) with outKB Identification** is the final stage. The objective is to determine the referent entity (or report *outKB* if none), given a table cell and its candidate entity set. The identification of outKB mentions significantly increases the complexity of the EL task, as it requires the method to differentiate between e.g. an unusual surface form of an inKB entity versus an outKB mention. However, distinguishing outKB mentions is a critical step in rapidly evolving domains like science, where existing KBs are highly incomplete.

## 3.2 Dataset Construction

Obtaining high-quality annotations for S2abEL is non-trivial. Identifying attributed sources and gold entities requires a global understanding of the text and tables in the full document. However, asking annotators to read every paper fully is prohibitively expensive. Presenting the full list of entities in the target KB to link from is also not feasible, while showing annotators short auto-populated candidate entity sets may introduce bias and miss gold entities. We address these challenges by designing a special-purpose annotation interface and pipeline, as detailed below.

In the construction process, we used two in-house annotators with backgrounds in data analytics and data science, both having extensive experience in reading and annotating scientific papers. In addition, one author of the paper (author A) led and initial training phase with the annotators, and another author of the paper (author B) was responsible for evaluating the inter-annotator agreement (IAA) at the end of the annotation process.

**Bootstrapping existing resources** — We began constructing our dataset by populating it with tables and cell type annotations from SegmentedTables[2] ([Kardas et al., 2020](#)), a dataset where table cells have been extracted from papers and stored in an array format. Each cell is annotated according to whether it is a paper, metric, and so on; and each paper is classified into one of eleven categories (e.g., NLI and Image Generation). To gather data for the ASM task, we fine-tuned a T5-small ([Raffel et al., 2022](#)) model to extract the last name of the first author, year, and title for each paper that appears in the reference section of any papers in our dataset from the raw reference strings. We then used the extracted information to search for matching papers in Semantic Scholar ([Kinney et al., 2023](#)), to obtain their abstracts. Since the search APIs do not always return the matching paper at the top of the results, we manually verified the output for each query.

**Target KB** — Papers with Code (PwC)[34] is a free and open knowledge base in the scientific domain with a total of 304,611 papers, 6,550 datasets, and 1,942 methods entities as of this writing. PwC includes basic relations between entities, such as relevant entities for a paper (denoted as *Paper-RelatesTo-Entity* in the rest of the paper), the introducing paper for an entity, etc. Its data is collected from previously curated results and collaboratively edited by the community. While the KB has good precision, its coverage is not exhaustive — in our experiments, 42.8% of our entity mentions are outKB, and many papers have empty Paper-RelatesTo-Entity relations.

**Human Annotation** — We developed a web interface using the Flask[5] library for the annotation process. It provides annotators with a link to the original paper, an indexed reference section, and annotation guidelines. The detailed annotation interface with instructions can be found at Appendix C.

For the CTC sub-task, we asked annotators to make necessary modifications to correct errors in SegmentedTables and accommodate the extra *dataset&metric* class. During this phase, 15% of the original labels were changed. For the ASM sub-task, annotators were asked to read relevant document sections for each cell and identify attributed sources, if any. This step can require a global understanding of the document, but candidate lists are relatively small since reference sections usually contain just tens of papers. For the EL sub-task, the web interface populates each cell with entity candidates that are 1) returned from PwC with the cell content as the search string, and/or 2) associated with the identified attributed paper(s) for this cell via the *Paper-RelatesTo-Entity* relation in PwC. Automatic candidate population is designed to be preliminary to prevent annotators from believing that gold entities should always come from the candidate set. Annotators were also asked to search against PwC using different surface forms of the cell content (e.g., full name, part of the cell content) before concluding that a cell refers to an outKB entity.

To ensure consistency and high quality, we conducted a training phase led by author A, where the two annotators were given four papers at a time to perform all annotation tasks. We then calculated the IAA between author A and each annotator for the four papers using Cohen's Kappa ([McHugh, 2012](#)), followed by disagreement discussion and guideline refinement. This process was repeated until the IAA score achieves "substantial agreement" as per ([McHugh, 2012](#)). Afterward, the remain-

---

[2][https://github.com/paperswithcode/axcell/releases](https://github.com/paperswithcode/axcell/releases)

[3]Our corpus is based on Papers with Code 2022/07 dump.

[4][https://github.com/paperswithcode/paperswithcode-data](https://github.com/paperswithcode/paperswithcode-data)

[5]flask.palletsprojects.com

|        | CTC    | ASM   | EL    |
|--------|--------|-------|-------|
| # papers | 327  | 316   | 303   |
| # tables | 886  | 790   | 732   |
| # cells | 52,257 | 9,564 | 8,429 |

Table 1: Overall statistics of S2abEL. It consists of 52,257 data points for cell types, 9,564 for attributed source matching, and 8,429 for entity linking, with ground truth.

ing set of papers was given to the annotators for annotation.

### 3.3 Dataset and Annotation Statistics

**Dataset Statistics** — Table 1 provides a summary of the statistics for S2abEL. ASM and EL annotations are only available for cells labeled positively in CTC. Metrics only are not linked to entities due to the lack of a controlled metric ontology in PwC. It is worth noting that S2abEL contains 3,610 outKB mentions versus 4,819 inKB mentions, presenting a significantly different challenge from prior datasets that mostly handle inKB mentions. More details are in Appendix A.

**Post-hoc IAA Evaluation** — We conducted a post-hoc evaluation to verify the quality of annotations, where author B, who is a researcher with a Ph.D. in Computer Science, independently annotated five random tables. The Cohen's Kappa scores show a substantial level of agreement (McHugh, 2012) between author B and the annotations (100% for CTC, 85.5% for ASM, and 60.6% for EL). These results demonstrate the quality and reliability of the annotations in S2abEL. A more detailed analysis on why EL agreement is relatively low can be found at Appendix D.

## 4 Method

In this section, we describe our approach for representing table cells, papers, and KB entities, as well as our model design for performing each of the sub-tasks defined in Section 3.1.

**Cell Representation** — For each table cell in a document, we collect information from both document text and the surrounding table. Top-ranked sentences were retrieved using BM25 (Robertson and Zaragoza, 2009) as context sentences, which often include explanations and descriptions of the table cell. The surrounding table captures the row and column context of the cell, which can offer valuable hints, such as the fact that mentions in the

same row and column usually refer to the same type of entities. More details about cell representation features are in Table 9.

**Paper Representation** — For each referenced paper, we extract its index in the reference section, the last name of the first author, year, title, and abstract. Index, author name, and year are helpful for identifying inline citations (which frequently take the form of the index in brackets or the author and year in parens). Additionally, the title and abstract provide a summary of a paper which may contain information on new concepts it proposes.

**KB Entity Representation** — To represent each entity in the target KB, we use its abbreviation, full name, and description from the KB, if available. The abbreviation and full name of an entity are crucial for capturing exact mentions in the text, while the description provides additional context for the entity (Logeswaran et al., 2019).

**Cell Type Classification** — We concatenate features of cell representation (separated by special tokens) and input the resulting sequence to the pre-trained language model SciBERT (Beltagy et al., 2019). For each token in the input sequence, we add its word embedding vector with an additional trainable embedding vector from a separate embedding layer to differentiate whether a token is in the cell, from context sentences, etc. (Subsequent mentions of SciBERT in this paper refer to this modified version). We pass the average of the output token embeddings at the last layer to a linear output layer and optimize for Cross Entropy loss. However, because the majority of cells in scientific tables pertain to experimental statistics, the distribution of cell types is highly imbalanced (as shown in Appendix A). To address this issue, we oversample the minority class data by randomly shuffling the context text extracted from the paper for those cells at a sentence level.

**Attributed Source Matching** — To enable contextualization between cell context and a potential source, we combine the representations of each table cell and potential attributed source in the document as the input to a SciBERT followed by a linear output layer. We optimize for the Binary Cross Entropy loss, where all non-attributed sources in the document are used as negative examples for a cell. The model output measures the likelihood that a source should be attributed to given a table cell.

**Candidate Entity Retrieval** — We design a method that combines candidates retrieved by two

strategies: (i) *dense retrieval (DR)* ([Karpukhin et al., 2020](#)) that leverages embeddings to represent latent semantics of table cells and entities, and (ii) *attributed source retrieval (ASR)* which uses the attributed source information to retrieve candidate entities.

For DR, we fine-tune a bi-encoder architecture ([Reimers and Gurevych, 2019](#)) with two separate SciBERTs to optimize a triplet objective function. The model is only trained on cells whose gold referent entity exists in the KB. Top-ranked most similar entities based on the BM25F algorithm ([Robertson and Zaragoza, 2009](#))[6] in Elasticsearch are used as negative examples. For each table cell $t_i$, the top-k nearest entities $\mathcal{O}_{dr}^i$ in the embedding space with ranks are returned as candidates.

For ASR, we use the trained ASM model to obtain a list of papers ranked by their probabilities of being the attributed source estimated by the model. The candidate entity sequence $\mathcal{O}_{asr}^i$ is constructed by fetching entities associated with each potentially attributed paper in ranked order using the Paper-RelatesTo-Entity relations in PwC. Only entities of the same cell type as identified in CTC are retained. Note that including entities associated with lower-ranked papers mitigates the errors propagated from the ASM model and the problem of imperfect entity and relation coverage that is common in real-world KBs.

We finally interleave $\mathcal{O}_{dr}^i$ and $\mathcal{O}_{asr}^i$ until we reach a pre-defined entity set size $K$.

**Entity Disambiguation with outKB Identification** — Given a table cell and its entity candidates, we fine-tune a cross-encoder architecture ([Reimers and Gurevych, 2019](#)) with a SciBERT that takes as input the fused cell representation and entity representation, followed by a linear output layer. We optimize for BCE loss using the same negative examples used in CER training. The trained model is used to estimate the probability that a table cell matches an entity. If the top-ranked entity for a cell has a matching likelihood lower than 0.5, then the cell is considered to be outKB.

## 5 Evaluations

As no existing baselines exist for the end-to-end table EL task with outKB mention identification, we compare our methods against appropriate recent

work by evaluating their performance on sub-tasks of our dataset (Section [5.1](#)). Additionally, we report the performance of the end-to-end system to provide baseline results for future work (Section [5.2](#)). Finally, to understand the connection and impact of each sub-task on the final EL performance, we conducted a component-wise ablation study (Section [5.3](#)). This study provides valuable insights into the difficulties and bottlenecks in model performance.

The experiments are designed to evaluate the performance of methods in a cross-domain setting (following the setup in [Kardas et al., 2020](#)), where training, validation, and test data come from different disjoint topics. This ensures that the methods are not overfitting to the particular characteristics of a topic and can generalize well to unseen data from different topics.

### 5.1 Evaluating Sub-tasks

#### 5.1.1 Cell Type Classification

We compare our method against AxCell's cell type classification component ([Kardas et al., 2020](#)), which uses a ULMFiT architecture ([Howard and Ruder, 2018](#)) with LSTM layers pre-trained on arXiv papers. It takes as input the contents of table cells with a set of hand-crafted features to provide the context of cells in the paper. We use their publicly available implementation[7] with a slight modification to the output layer to suit our 5-class classification.

Table [2](#) shows that our method outperforms AxCell somewhat in terms of F1 scores. Although we do not claim our method on this particular sub-task is substantially better, we provide baseline results using state-of-the-art transformer models.

#### 5.1.2 Candidate Entity Retrieval

Since the goal of CER is to generate a small list of potential entities for a table cell, we evaluate the performance of the CER method using *recall@K*.

Figure [2](#) shows the results of evaluating dense retrieval (DR), attributed source retrieval (ASR), and a combination of both methods, with different candidate size limits $K$. We observe that seeding the candidate set with entities associated with attributed papers significantly outperforms DR, while interleaving candidates from ASR and DR produces the most promising results. These results

---

[6]We chose to use BM25F instead of BM25 because it can take into account entity data with several fields, including name, full name, and description.

[7]https://github.com/paperswithcode/axcell

| | Validation | | | | | | Test | | | | | |
| Class | AxCell | | | Ours | | | AxCell | | | Ours | | |
| | P | R | $F_1$ | P | R | $F_1$ | P | R | $F_1$ | P | R | $F_1$ |
|---|---|---|---|---|---|---|---|---|---|---|---|---|
| *other* | 97.4 | 98.6 | 98.0 | 98.4 | 98.4 | 98.4 | 97.3 | 98.4 | 97.8 | 98.0 | 98.4 | 98.1 |
| *dataset* | 83.1 | 81.7 | 82.2 | 83.0 | 88.7 | 85.6 | 90.0 | 84.0 | 86.2 | 90.1 | 82.3 | 85.4 |
| *method* | 96.7 | 96.4 | 96.6 | 97.2 | 96.8 | 97.0 | 95.3 | 95.5 | 95.4 | 93.4 | 96.9 | 95.1 |
| *metric* | 71.3 | 72.8 | 71.9 | 88.6 | 79.7 | 83.7 | 86.6 | 86.1 | 85.0 | 88.0 | 87.4 | 86.8 |
| *dataset&metric* | 97.1 | 41.9 | 58.0 | 88.8 | 77.6 | 82.1 | 82.6 | 63.4 | 68.1 | 75.3 | 64.8 | 61.9 |
| Micro $F_1$ | | 95.8 | | | **96.8** | | | 96.0 | | | **96.2** | |

Table 2: Results of cell type classification on our method and AxCell, with image classification papers fixed as the validation set and papers from each remaining category as the test set in turn.

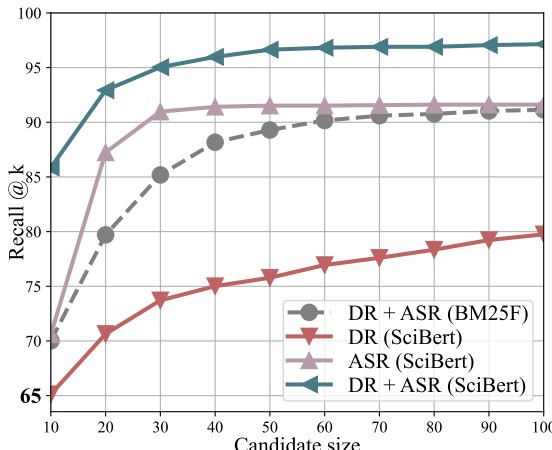

Figure 2: Evaluation of different candidate entity retrieval methods. The method in the parenthesis indicates whether a fine-tuned SciBERT or BM25F is used.

demonstrate the effectiveness of utilizing information on attributed sources to generate high-quality candidates. It is worth noting that when $K$ is sufficiently large, ASR considers all sources as attributed sources for a given cell, thus returning entities that are associated with any source. However, if the gold entity is not related to any cited source in the paper, it will still be missing from the candidate set. Increasing $K$ further will not recover this missing entity, as indicated by the saturation observed in Figure 2.

**Error Analysis** — We examined the outputs of ASR and identified two main challenges. First, we observed that in 22.8% of the error cases when $K = 100$, authors did not cite papers for referred concepts. These cases typically involve well-known entities such as LSTM (Hochreiter and Schmidhuber, 1997). In the remaining error cases, the authors did cite papers; however, the gold entity was not retrieved due to incomplete *Paper-RelatesTo-Entity* relations in the target KB

or because the authors cited the wrong paper.

We additionally investigated the error cases from DR and found that a considerable fraction was caused by the use of generic words to refer to a specific entity. For instance, the validation set of a specific dataset entity was referred to as "val" in the table, the method proposed in the paper was referred to as "ours", and a subset of a dataset that represents data belonging to one of the classification categories was referred to as "window". Resolving the ambiguity of such references requires the model to have an understanding of the unique meaning of those words in the context.

When using the combined candidate sets, missing gold entities were only observed when both DR and ASR failed, leading to superior performance compared to using either method alone.

### 5.1.3 Entity Disambiguation with inKB Mentions

The state-of-the-art method closest to our table EL task is TURL (Deng et al., 2020), designed for general-domain tables with inKB cells. It is a structure-aware Transformer encoder pre-trained on the general-purpose WikiTables corpus (Bhagavatula et al., 2015), which produces contextualized embeddings for table cells, rows, and columns that are suitable for a range of downstream applications, including table EL. We used TURL's public code[8] and fine-tuned it on the inKB cells of our dataset and compared it with our method using the same entity candidate set of size 50.

Table 3 shows that our model achieves a substantial improvement in accuracy over TURL on nine out of ten paper folds. The examples in Table 6 (appendix) demonstrate that our model is more effective at recognizing the referent entity when the cell mention is ambiguous and looks similar to other en-

---

[8] https://github.com/sunlab-osu/TURL

| Test fold | Support | TURL | Ours |
|---|---|---|---|
| Question ans. | 381 | 15.0 | **36.1** |
| Object det. | 2040 | 34.1 | **41.0** |
| Speech rec. | 175 | 34.3 | **54.3** |
| Image gen. | 168 | 7.7 | **35.1** |
| Machine trans. | 234 | 15.4 | **39.3** |
| Text class. | 246 | 52.4 | **68.7** |
| Pose estim. | 108 | 36.1 | 36.1 |
| Semantic seg. | 641 | 44.8 | **50.9** |
| NLI | 328 | 30.8 | **58.8** |
| Misc. | 81 | 14.8 | **33.3** |
| Micro avg | | 32.5 | **44.8** |

Table 3: Accuracy for end-to-end entity linking for cells that refer to an inKB entity with 10-fold-cross-domain evaluation using our approach and TURL. Our method is specialized for tables in scientific papers and outperforms the more general-purpose TURL method.

| Test fold | O/I ratio | OutKB $F_1$ | InKB hit@1 | Overall acc. |
|---|---|---|---|---|
| Machine trans. | 0.60 | 62.2 | 23.7 | 50.3 |
| Image gen. | 0.48 | 55.1 | 20.0 | 44.4 |
| Misc. | 2.74 | 85.1 | 19.5 | 74.9 |
| Speech rec. | 1.46 | 73.7 | 33.3 | 66.5 |
| Question ans. | 2.33 | 84.2 | 14.0 | 69.3 |
| NLI | 1.11 | 80.6 | 47.4 | 68.3 |
| Text class. | 1.24 | 77.1 | 35.4 | 66.8 |
| Object det. | 0.23 | 49.6 | 35.2 | 45.7 |
| Semantic seg. | 0.42 | 73.4 | 39.7 | 55.0 |
| Pose estim. | 1.06 | 72.2 | 35.2 | 59.9 |
| Micro avg | 0.75 | 71.4 | 33.3 | 57.6 |
| + gold CTC | 0.75 | 72.4 | 33.4 | 58.2 |
| + gold Can. | 0.75 | 71.5 | 33.4 | 57.6 |
| + gold both | 0.75 | 72.5 | 33.4 | 58.2 |

Table 4: End-to-end EL results with 10-fold-cross-domain evaluation of our method on learned DR + ASR candidate sets of size 50 with the inKB threshold set to 0.5. Although our model achieved reasonable overall accuracy, it is still far from perfect, leaving ample room for future improvements in the end-to-end table EL task.

tities in the KB. This is because TURL as a generic table embedding method focuses on just cell content and position while our approach combines cell content with the full document. Our analysis further reveals that TURL made incorrect predictions for all cells whose mentions were shorter than four characters (likely an abbreviation or a pointer to a reference paper). Meanwhile, our method correctly linked 39% of these cells.

## 5.2 End-to-end Evaluation

We now evaluate the end-to-end performance of our approach on the EL task with outKB identification. In addition to re-ranking candidate en-

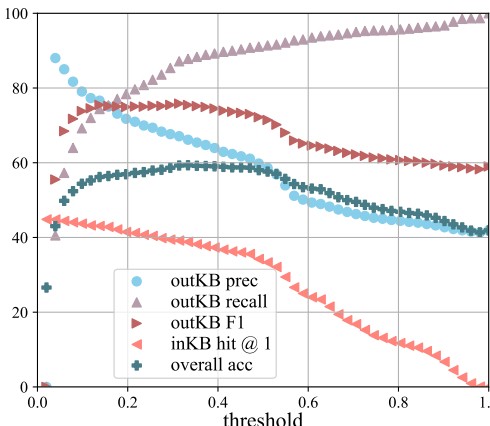

Figure 3: Entity linking results with varying inKB thresholds. Note that the inKB hit rate is low (44.8%) even when all mentions are predicted with an entity (i.e., threshold is 0).

tities, the method needs to determine when cell mentions refer to entities that do not exist in the target KB. We report $F_1$ scores for outKB entities as the prediction is binary (precision and recall are reported in Appendix Table 8). For inKB mentions, we report the hit rate at top-1. Additionally, we evaluate overall performance using accuracy.[9] For each topic of papers, we report the ratio of outKB mentions to inKB mentions. The top block of Table 4 shows the end-to-end EL performance of our method. Our analysis shows a positive Pearson correlation (Cohen et al., 2009) of 0.87 between O/I ratio and overall accuracy, indicating our method tends to higher accuracy on datasets with more outKB mentions. Figure 3 shows the performance at various inKB thresholds.

**Error Analysis** — We sampled 100 examples of incorrect predictions for both outKB and inKB mentions and analyzed their causes of errors in Table 7 (Appendix E). Our analysis reveals that a majority of incorrect inKB predictions are due to the use of generic words. For outKB mentions, the model tends to get confused when they are similar to existing entities in the target KB.

## 5.3 Component-wise Ablation Study

To investigate how much of the error in our end-to-end three-step system was due to errors introduced in the first two stages (specifically, wrong cell type

---

[9]A cell is considered a correct prediction if it is an outKB mention and predicted as such, or if it is an inKB mention and predicted as inKB with the gold entity being ranked at top 1.

classifications from CTC or missing correct candidates from CER), we tried measuring system performance with these errors removed. Specifically, we tried replacing the CTC output with the gold cell labels, or adding the gold entity to the output CER candidate set, or both.

The results in the bottom block of Table 4 show that there is no significant difference in performance with gold inputs. This could be because CTC and CER are easier tasks compared to ED, and if the model fails those tasks, it is likely to still struggle to identify the correct referent entity, even if that is present in the candidate set or the correct cell type is given.

## 6 Conclusion

In this paper, we present S2abEL, a high-quality human-annotated dataset for Entity Linking in machine learning results tables, which is, to the best of our knowledge, the first dataset for table EL in the scientific domain. We propose a model that serves as a strong baseline for the end-to-end table EL task with identification of outKB mentions, as well as for each of the sub-tasks in the dataset. We show that extracting context from paper text gives a significant improvement compared to methods that use only tables. Identifying attributed source papers for a concept achieves higher recall@k compared with dense retrieval for candidate entity generation. However, the best baselines still fall far below human performance, showing potential for future improvement.

## Limitations

In this section, we discuss some limitations of our work. First, our dataset only includes tables from English-language papers in the machine learning domain, linked to the Papers with Code KB, which limits its generalizability to other domains, languages, and KBs. Second, we acknowledge that the creation of S2abEL required significant manual effort from domain experts, making it a resource-intensive process that may not be easily scalable. Third, our approach of using attributed papers to aid in identifying referent entities relies on the target KB containing relations that associate relevant papers and entities together. Fourth, we do not compare against large GPT-series models and leave this as future work. Finally, while our experiments set one initial baseline for model performance on our task, substantially more exploration of different methods may improve performance on our task substantially.

## Acknowledgments

This work was supported in part by NSF Grant 2033558.

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

| Fold | CTC | | | ASM | | | EL | | |
|---|---|---|---|---|---|---|---|---|---|
| | # paper | # table | # cell | # paper | # table | # cell | # paper | # table | # cell |
| Question ans. | 58 | 139 | 6,422 | 57 | 128 | 1,320 | 57 | 127 | 1,282 |
| Object det. | 54 | 159 | 17,020 | 52 | 141 | 2,686 | 51 | 135 | 2,527 |
| Image class. | 27 | 94 | 2,681 | 27 | 82 | 608 | 27 | 81 | 597 |
| Speech rec. | 22 | 88 | 3,516 | 21 | 76 | 649 | 21 | 74 | 612 |
| Image gen. | 25 | 37 | 1,184 | 23 | 34 | 290 | 23 | 34 | 288 |
| Machine trans. | 28 | 48 | 2,199 | 25 | 42 | 412 | 25 | 41 | 378 |
| Text class. | 21 | 75 | 4,085 | 20 | 55 | 688 | 19 | 51 | 600 |
| NLI | 32 | 83 | 3,385 | 30 | 68 | 787 | 30 | 66 | 697 |
| Pose estim. | 13 | 47 | 4,447 | 11 | 31 | 550 | 7 | 18 | 222 |
| Semantic seg, | 32 | 82 | 5,733 | 30 | 75 | 927 | 30 | 75 | 919 |
| Misc. | 15 | 34 | 1,585 | 13 | 30 | 308 | 13 | 30 | 307 |

Table 5: Detailed statistics of S2abEL.

| Cell Content | Column header | TURL result | Our result | Gold entity |
|---|---|---|---|---|
| | | InKB | | |
| "[33]" | "" | Cityscapes, PoseTrack, LAMBADA | PointNet, GAN, CRF | PointNet |
| "Text GCN" | "Model" | Global Conv. Net., End-to-End Mem. Net. | Graph Conv. Net., Global Conv. Net. | Graph Conv. Net. |

Table 6: Incorrect examples for end-to-end EL from TURL. The table includes the cell content and the column header in the first two columns, the top-3 ranked results from TURL and our approach in the third and fourth columns, respectively, and the gold entity in the last column.

Shuo Zhang, Edgar Meij, Krisztian Balog, and Ridho Reinanda. 2020. Novel entity discovery from web tables. In *Proceedings of The Web Conference 2020*, WWW '20, page 1298–1308, New York, NY, USA. Association for Computing Machinery.

Susan Zhang, Stephen Roller, Naman Goyal, Mikel Artetxe, Moya Chen, Shuohui Chen, Christopher Dewan, Mona Diab, Xian Li, Xi Victoria Lin, et al. 2022. Opt: Open pre-trained transformer language models. *arXiv preprint arXiv:2205.01068*.

## A  Detailed Dataset Statistics

S2abEL consists of 11 folds, each corresponding to a topic. Table 5 provides detailed statistics on the number of papers, tables, and cells for each sub-task and topic. The class distribution for CTC is as follows: *other* (74%), *dataset* (8%), *method* (14%), *metric* (3%), and *dataset&metric* (0.4%). For ASM, 1,532 (16.6%) cells have missing attributed paper, 1,095 (11.9%) cells attribute to the paper itself, 6,598 (71.5%) cells attribute to an entry in the reference section of the paper. For EL, 3,610 (42.8%) cells refer to outKB entities and (57.2%) cells refer to inKB entities.

## B  Training Details

We trained all our models for two epochs with a batch size of 32, using the AdamW optimizer (Loshchilov and Hutter, 2019) with linear decay warm-up. The initial learning rate was 2e-5 and the warm-up ratio was 10%. All models were trained using a single 48Gb NVIDIA A6000 GPU. For the triplet loss function in DR, we used Euclidean as the distance function with a margin of 1. For the Candidate Entity Retrieval and Entity Disambiguation tasks, we used negative examples of size 50 at training time. Additionally for the ED task, we set candidate set size limitation as 50 when making predictions.

## C  Annotation Interface and Guidelines

Our annotation interface with annotation guidelines is at https://github.com/allenai/s2abel/blob/main/common_utils/Annotation%20Interface.pdf. Note that there might be cells that contain a subentity mentions consisting of an entity mention and a non-entity mention string, e.g., "Bert-large", "Bert

| Cause | Percent | Example cell | Our result | Gold entity |
|---|---|---|---|---|
| | | | InKB | |
| Variants | 21% | "bottle" | PASCAL VOC, PASCAL VOC 2007 | PASCAL VOC 2007 |
| | | "BigGAN" | BigGan-deep, BigGan | BigGan |
| Top below threshold | 16% | "Car" | *outKB*, KITTI | KITTI |
| Abbreviations | 19% | "FR-EN" | Arcade Learning Environment | WMT 2014 |
| Generic words | 39% | "Ours" | Neural Turing Machine | OHEM |
| | | | OutKB | |
| Similar names | 56% | "DPN" | Dual Path Network | Deep Parsing Network |
| Generic words | 22% | "12" | HUB5 English | bAbi |

Table 7: Representative examples of erroneous end-to-end EL cases. The table includes the cause and the percentage for that cause in the first two columns, an example of cell content for that cause and our incorrect prediction in the third and fourth columns, and the gold entity in the last column.

| Test fold | Precision | Recall |
|---|---|---|
| Machine trans. | 46.4 | 94.4 |
| Image gen. | 38.7 | 95.7 |
| Misc. | 77.0 | 95.1 |
| Speech rec. | 62.8 | 89.3 |
| Question ans. | 76.9 | 93.1 |
| NLI | 74.9 | 87.2 |
| Text class. | 66.2 | 92.2 |
| Object det. | 34.0 | 91.1 |
| Semantic seg. | 61.4 | 91.2 |
| Pose estim. | 63.8 | 83.3 |
| Micro avg | 58.6 | 91.4 |
| + gold CTC | 59.5 | 92.7 |
| + gold Can. | 58.7 | 91.4 |
| + gold both | 59.5 | 92.7 |

Table 8: Additional end-to-end Entity linking results for outKB cells.

| Feature | Description |
|---|---|
| cell content | cell's raw text |
| region | cell's relative location with reference to the top-left numeric cell in the table, i.e., top-left, top-right, bottom-left, and bottom-right |
| context sentences | top-ranked sentences in the full document (including table captions, section headers, etc.) regarding the cell content based on BM25 |
| row context | concatenated cell's row separated by special tokens |
| column context | concatenated cell's column separated by special tokens |
| position | cell's 2D position in the table in terms of distance from the top left corner |
| reverse position | cell's 2D position in the table in terms of distance from the bottom right corner. |
| has reference | whether cell has a reference |

Table 9: Features for cell representation.

## D   Annotation Error Analysis

Our investigation showed that the main cause of disagreement in the EL phase was that there were cells whose matching entities were confusing to the annotators, due to their insufficient background in the specific academic area and/or the paper not clearly indicating which entity it was and/or whether should be considered a variant of an existing entity or a different entity entirely.

## E   Error Case Study

Table 6 presents examples where TURL made incorrect EL predictions while our approach made correct predictions. Table 7 summarizes the main causes of incorrect predictions made by our approach for both inKB and outKB mentions.

with 6 layers frozen". For these cells, we asked the annotators to focus on the primary entity and our current model considers these mentions as mentions of the main entity. Thus those two mentions are labeled as *method*, and linked to https://paperswithcode.com/method/bert. We also specifically asked the annotators to mark cells that contain mentions of more than one primary entity or are confusing to understand, which are excluded from the dataset. We leave the tasks of linking subentities explicitly and cells to multiple entities for future work.