# OpenReview forum: "S2abEL: A Dataset for Entity Linking from Scientific Tables"
_EMNLP/2023/Conference — EMNLP 2023 Main_

### Official Review · Reviewer_CtHC · 2023-07-31

**Soundness:** 2

**Excitement:**

3: Ambivalent: It has merits (e.g., it reports state-of-the-art results, the idea is nice), but there are key weaknesses (e.g., it describes incremental work), and it can significantly benefit from another round of revision. However, I won't object to accepting it if my co-reviewers champion it.

**Paper Topic And Main Contributions:**

This paper presents a new manually annotated dataset for the entity linking task in tables and a new method for automatically detecting and linking table cells. The task is dedicated to machine learning experimental results reported in tables. The entity to be linked are those that are defined in Paper with code website, other relevant entities are considered as out-of-knowledge-base. The dataset is annotated by two annotators in a single way with double-blind on a limited subset. The entity linking method is divided into four successive steps, namely extraction and typing of the mentions, identification of the reference paper for the mentions if relevant, and filtering a subset of mentions that are good candidates for the last entity linking step. The end-to-end method achieves better results than the general method TURL.

**Questions For The Authors:**

A.	Are the internal cells - usually the numerical values of the metrics for a given method and dataset - extracted and linked?

**Reasons To Accept:**

The aim of the work, i.e. to extract information from experimental result tables, is relevant because comparing methods on reference datasets has become a very important part of machine learning research. Linking the methods and datasets cited in the tables to the primary source of information is an important step towards reproducibility and fair comparison. Papers with code website is becoming a popular source of information for this comparison. The table mentions are highly ambiguous, mostly acronyms or abbreviations that require external text analysis to disambiguate. The dataset and the method proposed to solve this problem are relevant. The experimental protocol is detailed.

**Reasons To Reject:**

The main issue is the lack of clarity on the precise objective of the task. The first point is related to the cells of the table that are effectively considered. All the examples are about the names of columns and rows, and the entity types are method, dataset, metric, and dataset&metric. It is not clear at all if the internal cells, usually the numerical values of the metrics for a given method and dataset, are also extracted and linked. Column and row names should absolutely not be called cell content if internal cells are not processed. If they are, the way the right column and row are detected from the table structure, and the way the values are obtained should be discussed (taken from the primary source cited, or reproduced by the authors). The entities of the metrics type are not normalized with the KB (line 331), this should be mentioned in the task definition section.
The second major problem relates to the reference KB the description of which is seriously insufficient for the reader to understand the aim of the work, its challenges, and limitations. The KB components, data types, relationships, and content must be formally described and related to the table content. The generic URL of the paperwithcode website is not enough. The canonicalized ontology (line 158) and the taxonomy (line 162) must be defined. The “topics” (section 5, line 474 and following) also play an important role in the method. The paper must explain how they are extracted from the KB, and whether they are a subset of the KB topics. Their full names should be given. It is not sufficient to list their abbreviation in Table 5. The relationships of the KB are mentioned as helpful to the method, but it is unclear what the relationships are, and how they are used by the method.
The lack of discussion on the structure of the table is a third shortcoming. The reader understands that the column and row names are extracted and linked to the KB independently of each other. Besides internal cell values, which are obviously dependent in 2-D table of the row and column meaning, row and column meaning is also dependent on their context within the table when tables are not just in the form of a simple grid. There are three levels of column names in Table 2 as an example. The way complex structures are taken into account by the EL method should be discussed.
Less seriously, the low rate of inter-annotator agreement for EL (66%) is below the usual standard for this type of corpus. It is presented as satisfactory. The paper should discuss and analyze the reason for this, which may be due to the guidelines, or the lack of expertise of the annotators. Since the reader assumes that the paper content is not ambiguous, could the annotation difficulty be due to the KB, which could contain several possible references for the same entity?
The authors emphasize the ability of the STabEL method to detect out-of-the-knowledge-base entities compared to other methods that would not be able to do it. The authors seem to confuse about what is due to the closed-world assumption (KB incompleteness) and what is due to the KB structure. Usually, in information extraction, the entity is recognized together with its type. Depending on the tasks, the reference KB for a given type can be a flat non-hierarchical nomenclature where entity-linking consists of assigning a nomenclature entry to the entity whether there exists one. The reader understands that the scientific table entity linking task here belongs to this family. This is not more realistic or less realistic in scientific domains (Lines 54-60, lines 116-129). Many reference KB for a given type are hierarchical (diseases, anatomy parts, chemical compounds), and the entity-linking tasks consist of identifying the most relevant node, possibly general, which is much more relevant than considering the entity as out-of-knowledge base. Without judging the value of others, the authors should simply describe their own objective, EL with “flat” KB.


**Reproducibility:**

1: Could not reproduce the results here no matter how hard they tried.

**Reviewer Confidence:**

4: Quite sure. I tried to check the important points carefully. It's unlikely, though conceivable, that I missed something that should affect my ratings.

**Typos Grammar Style And Presentation Improvements:**

The goal of Section 2.2 Scientific IE could be revised to focus on the paper subject, i.e. scientific table, or ML method or dataset name extraction, entity-linking to KB, and remove more general references that are either not relevant or incomplete.
Line 287, the URL of the guidelines must be given.
Line 320, the number of articles must be given.
Line 364, what is Appendix 9?
Lines 397-398, please explain how oversampling is achieved by shuffling the sentences.
Line 453, the 0.5 threshold should be experimentally confirmed.
The second sentence of the Table 2 legend is redundant to the text and should be removed.
Line 518 Have all errors been examined?
Line 624 The expression “may limit its generalizability” is an understatement.

---

> ### Author Rebuttal · Authors · 2023-08-28
>
> Thank you for the very detailed feedback. We acknowledge the concerns raised and aim to address each one.
>
> 1. **Internal cells**: We consider all cells of a table in our task, as sourced from the SegmentedTable dataset from Kardas et al., (2020) (Lines 257-258).  This dataset extracts tables into two-dimensional arrays, and each cell is annotated according to whether it is a dataset, metric, and so on (Lines 258-260). This dataset also classifies the topic of a paper into one of eleven categories (e.g, NLI and Image Generation). In our Cell Type Classification sub-task, cells that do not refer to datasets, metrics, or methods are identified (e.g., the numeric value cells). These cells, thus, are not considered in subsequent Entity Linking steps (Lines 175-179).
> 2. **Knowledge Base (KB) description**:  We concur with your observation about the need for a clearer presentation of the KB. We'll enhance this description in the revised manuscript. We believe this is a beneficial presentation improvement, but not a technical issue. There was also some confusion around the “topics”. We used the topic classification as it is from the SegmentedTable dataset.
> 3. **Discussion of table structure**: We used tables extracted in the SegmentedTable dataset from Kardas et al., (2020) (Lines 257-258). We acknowledge the presence of cells with diverse grid structures and have used them as parsed by the SegmentedTable, including their approach to multicolumn cells. We will add further discussion of this processing in our revisions.
> 4. **IAA score**: We'd like to highlight that a Cohen Kappa score in the 0.61–0.80 range is deemed as "substantial agreement" as per McHugh (2012). We stopped fine-tuning the guidelines once this agreement level was achieved. The location of the guideline is given on Line 954.  Our investigation showed that the main cause of disagreement in the EL phase was that there were cells whose matching entities were confusing to the annotators, due to their insufficient background in the specific academic area and/or the paper not clearly indicating which entity it was and/or whether should be considered a variant of an existing entity or a different entity entirely. We will add a discussion of this in the paper or its appendix.
> 5. **STabEL Method & Out-of-KB Entities**: We appreciate your insight into the distinction between closed-world assumptions and KB structure. We'll ensure the manuscript accurately represents our non-hierarchical "flat" KB and provides a more fair and balanced discussion on different types of Entity Linking.
> 6. **Reproducibility**: We have provided as supplementary material our dataset along with the code to reproduce our main results. If there are any other specific reproducibility concerns leading to the current score (1), we would be very happy to address them.
>
> Lastly, we genuinely appreciate the suggestions in the “Typos Grammar Style And Presentation Improvements” section. These insights will guide us in refining the manuscript.
>
> Thank you for your time and consideration.
>
> References:
> Kardas et al., (2020), AXCELL: Automatic Extraction of Results from Machine Learning Papers
> McHugh (2012), Interrater reliability: the kappa statistic

---

### Official Review · Reviewer_zr1c · 2023-08-05

**Soundness:** 4

**Excitement:**

3: Ambivalent: It has merits (e.g., it reports state-of-the-art results, the idea is nice), but there are key weaknesses (e.g., it describes incremental work), and it can significantly benefit from another round of revision. However, I won't object to accepting it if my co-reviewers champion it.

**Paper Topic And Main Contributions:**

This paper presents a human-labeled dataset for entity linking in machine-learning result tables and a corresponding pipeline neural baseline. The unique feature of this dataset is that it explicitly addresses the problem of "outKB mentions", which refers to entities not yet in the reference KB.

**Questions For The Authors:**

1. What is the "Paper-RelatesTo-Entity relation in PwC" (lines 303-304)?

**Reasons To Accept:**

1. The paper is well-written and easy to follow. The authors clearly describe the dataset construction process and the model details.
2. The dataset is useful for the community and set a good example for future dataset construction on similar tasks but different domains.
3. The authors provide a reasonable baseline for the proposed dataset and conduct a thorough analysis of experimented methods.

**Reasons To Reject:**

The biggest weakness of this paper for me is that it does not clearly discuss its relation to the previous work of Kardas et al., (2020). In Section 2.2 (lines 144-148), the authors describe Kardas et al., (2020) as "However, they only identify entities as raw strings extracted from the set of papers they examine, without ... actual linking to an external", which is not the case. In fact, Kardas et al., (2020) also present a similar linking framework as the proposed one in this paper (linking cells to leaderboards/KB) while focusing on the result tuples (task, dataset, metric, score) instead of the entities. To me, this paper is more like an extension of Kardas et al., (2020) to the entity level with more fine-grained sub-task definitions, e.g., the inclusion of Attributed Source Matching and Candidate Entity Retrieval, and showing more recent Transformer models' performance. The proposed dataset is constructed upon the SegmentedTables dataset proposed by Kardas et al., (2020). Moreover, the AxCell model proposed by Karas et al., (2020) can be potentially used as an end-to-end baseline for this paper, where outKB mentions are entities that have low linking scores (below a threshold).

Reference:

Kardas et al., (2020), AXCELL: Automatic Extraction of Results from Machine Learning Papers

**Reproducibility:**

4: Could mostly reproduce the results, but there may be some variation because of sample variance or minor variations in their interpretation of the protocol or method.

**Reviewer Confidence:**

4: Quite sure. I tried to check the important points carefully. It's unlikely, though conceivable, that I missed something that should affect my ratings.

---

> ### Author Rebuttal · Authors · 2023-08-28
>
> Thank you for your thoughtful review and pointing out the lack of clarity regarding the work of Kardas et al., (2020). We acknowledge that our description of Kardas et al., (2020) and the comparison of our work and Kardas et al., in Section 2.2 was not sufficiently detailed. We do draw upon their work significantly in ours.
>
> Here's a clearer comparison between our work and that of Kardas et al., which we will incorporate in our revisions:
> 1. **Task relation with Kardas et al., (2020)**: Given a set of papers, Kardas and team begin by manually extracting a taxonomy of tasks, datasets, and metric names from those papers. This is the most important distinction from our work: whereas our data set maps each entity to an existing canonical external KB of models and datasets (PwC), they target a taxonomy manually built from the particular papers and surface forms they extract from. We would expect that the lexical similarity between the paper surface forms and the taxonomy entity names is thus higher than it would be for an external KB.  The AxCell tool subsequently maps corresponding numeric cells to the extracted taxonomy entries under a closed-world assumption. Notably, this taxonomy emphasizes lexical representations, with entities such as "AP" and "Average Precision" treated as distinct entities in the taxonomy despite being identical metrics in reality, due to different surface forms appearing in the papers. This is evident from their released taxonomy ( https://github.com/paperswithcode/axcell/releases/download/v1.0/models.tar.xz).
> Contrasting this with our approach:
>      1. Surface Form Extraction: The entity linker in Kardas et al. is based on lexical features, making it adept for their specific taxonomy. However, when aiming for Entity Linking to a canonical Knowledge Base (KB), such a method struggles due to variations in entity mentions, like "USA", "U.S.", and "America" all denoting the same entity.  Our dataset, by contrast, resolves different ambiguous surface forms from papers to their referent entities in te PWC knowledge base.
>      2. Entity Deduplication and Clarity: The taxonomy in Kardas et al. lacks robust deduplication or clustering of identical entities across papers and does not provide additional information other than names, resulting in potential ambiguities.
> 2. **Using the linker in Karadas et al., (2020) in our task**: The entity linker in Kardas et al., (2020) extracts lexical features of a mention in a cell, including Bag-of-Words and abbreviation extension and calculates the probability of matching to a (task, dataset, metric) tiple similar to TF-IDF, with 18 manually-tuned hyperparameters in the formula (as seen in Table 9 of their paper). Our initial research suggested that KB Entity Linking demands deeper linguistic and semantic features, which a predominantly lexical linker does not effectively capture. Moreover, the choice of the 18 parameters in Kardas et al., remains opaque, making replication or adaptation to new tasks challenging.
>
> Regarding your question on the "Paper-RelatesTo-Entity relation in PwC (lines 303-304): The Papers with Code KB provides methods and datasets that are relevant to a paper (e.g., Methods and Datasets sections on this page: https://paperswithcode.com/paper/code-llama-open-foundation-models-for-code). However, there is no guarantee of coverage or correctness of this feature. In fact, we found a significant number of papers that have incomplete or even completely blank Methods and Dataset sections in PwC. Due to this limitation, it became necessary to incorporate candidate entities from both direct dense retrieval and attributed source matching, as discussed in Section 4, line 410.

---

### Official Review · Reviewer_NRhP · 2023-08-06

**Soundness:** 4

**Excitement:**

4: Strong: This paper deepens the understanding of some phenomenon or lowers the barriers to an existing research direction.

**Paper Topic And Main Contributions:**

This paper presents a new dataset for entity linking in scientific tables. The entity linking task studied in the paper is to link entities in tables from scientific papers to entities in the Papers with Code taxonomy. To build the dataset, a special-purpose annotation interface and pipeline is developed. The paper also proposes a baseline model for the EL task.

**Reasons To Accept:**

1.A new dataset for EL in scientific tables is presented. The new dataset is large-scale and can be used for training and testing models on table EL.

2.A baseline model for table EL in scientific papers is proposed, which achieves good results in the experiments.

3.The paper is well structured and written.



**Reasons To Reject:**

As discussed in Limitations, only papers in the machine learning domain are used to built the dataset. Papers in different domains may have different patterns in tables. Papers from more domains need to be included in the dataset.

**Reproducibility:**

4: Could mostly reproduce the results, but there may be some variation because of sample variance or minor variations in their interpretation of the protocol or method.

**Reviewer Confidence:**

3: Pretty sure, but there's a chance I missed something. Although I have a good feel for this area in general, I did not carefully check the paper's details, e.g., the math, experimental design, or novelty.

---

> ### Author Rebuttal · Authors · 2023-08-28
>
> Thank you for highlighting this limitation in our paper.
>
> We also mentioned the same point in our Limitations section (Lines 621-625). We totally agree with the importance of diversifying the dataset to capture broader patterns in other domains, and this is one of the future steps.
>
> However, we would argue that our dataset nonetheless forms a significant contribution to the community as it is the only existing scientific table entity linking dataset with out-of-KB mentions, and it covers a broad range of the large ML field (spanning eleven different sub-disciplines).

---

### Meta-Review · Area_Chair_rkBu · 2023-09-16

**Recommendation:** 5

**Metareview:**

The paper presents a new dataset for entity linking in scientific tables - a dataset of tables are annotated with links to the papers with code taxonomy using a custom annotation interface.  It appears this is the first work to construct such an entity linking dataset for tables in scientific papers.

As the authors discuss in the limitations section, the dataset only covers one domain (results tables in machine learning papers).  It would be nice to see more domains, but this still seems like an important contribution.

---

### Decision · Program_Chairs · 2023-10-07

**Decision:**

Accept-Main

**Comment:**

The paper presents a new dataset for entity linking in scientific tables - a dataset of tables are annotated with links to the papers with code taxonomy using a custom annotation interface.  It appears this is the first work to construct such an entity linking dataset for tables in scientific papers.

As the authors discuss in the limitations section, the dataset only covers one domain (results tables in machine learning papers).  It would be nice to see more domains, but this still seems like an important contribution.